# Efficacy and safety of neoadjuvant stereotactic body radiation therapy plus dalpiciclib and exemestane for hormone receptor-positive, HER2-negative breast cancer: A prospective pilot study

Yu Zhang[1,2,3,4†], Shuo Cao[1†], Nan Niu[1,2,3†], Huilian Shan[1†], Jinqi Xue[1,2,3†], Guanglei Chen[1,2,3], Yongqing Xu[1], Jianqiao Yin[1], Chao Liu[1,2,3], Lisha Sun[1,2,3], Xiaofan Jiang[1,2,3], Meiyue Tang[1], Qianshi Xu[1,2], Mingxuan Jia[1], Xu Zhang[1], Zhenyong Zhang[1], Qingfu Zhang[5], Jianfei Wang[6], Ailin Li[1], Yongliang Yang[7*], Caigang Liu[1,2,3*]

[1]Department of Oncology, Shengjing Hospital of China Medical University, Shenyang, China; [2]Cancer Stem Cell and Translational Medicine Laboratory, Shengjing Hospital of China Medical University, Shenyang, China; [3]Innovative Cancer Drug Research and Development Engineering Center of Liaoning Province, Shenyang, China; [4]Department of Gastrointestinal Surgery, Yantai Affiliated Hospital of Binzhou Medical University, Yantai, China; [5]Department of Pathology, the First Affiliated Hospital and College of Basic Medical Sciences of China Medical University, Shenyang, China; [6]Jiangsu Hengrui Pharmaceuticals, Shanghai, China; [7]School of Bioengineering, Dalian University of Technology, Dalian, China

*For correspondence:
everbright99@163.com (YY);
angel-s205@163.com (CL)

†These authors contributed equally to this work

## eLife Assessment

The study evaluates the feasibility, safety, and tolerability of neoadjuvant radiotherapy followed by a CDK4/6 inhibitor (dalpiciclib) and hormonal therapy in treatment-naive patients with unilateral early-stage HR+/HER2- breast cancer. The findings are **convincing**, with a strong scientific rationale supported by integrated correlative studies. The trial is considered to be **important** as the outcomes could inform the design of larger, future studies. The limitations of the study have been acknowledged and outlined in this manuscript, which include only a small cohort of patients (n=12), which was not adequately powered to definitively assess the efficacy or safety of this combinatorial treatment approach.

## Abstract

**Background:** Both neoadjuvant chemotherapy and endocrine therapy only result in trivial pathological complete response rates and moderate objective response rates (ORR) in hormone receptor (HR)-positive, human epidermal growth factor receptor-2 (HER2)-negative breast cancer, more promising alternatives are urgently needed. With proven synergistic effect of cyclin-dependent kinase 4/6 (CDK4/6) inhibitor and radiotherapy in preclinical studies, this pilot study aimed to explore the efficacy and safety of neoadjuvant stereotactic body radiation therapy (SBRT) followed by dalpiciclib and exemestane in HR-positive, HER2-negative breast cancer.

**Methods:** This was a single-arm, non-controlled prospective pilot study. Treatment-naive patients with unilateral HR-positive, HER2-negative breast cancer received neoadjuvant radiotherapy (24 Gy/3 F) followed by dalpiciclib and exemestane for six cycles. The primary endpoint was the proportion of patients with residual cancer burden (RCB) score of 0-I. Key secondary endpoints included ORR, breast-conservation rate, biomarker analysis, and safety.

**Results:** All 12 enrolled patients completed the study treatment and surgery. Two (16.7%) of them achieved the RCB 0-I with the ORR of 91.7% (11/12). Analyses of tumor specimens showed significant increase of infiltrating T cells rather than alteration of PD-L1 positive immune cells. The most common grade 3 adverse events (AEs) were neutropenia (66.7%) and leukopenia (25.0%), but no grade 4–5 AE or death occurred.

**Conclusions:** Our results suggested neoadjuvant SBRT followed by dalpiciclib and exemestane is effective and tolerable and provides novel insights for the neoadjuvant treatment of HR+/HER2- breast cancer, which may be considered as a feasible option for patients with HR-positive, HER2-negative breast cancer.

**Funding:** None.

**Clinical trial number:** ClinicalTrials.gov: NCT05132790

## Introduction

Hormone receptor (HR)-positive, human epidermal growth factor receptor 2 (HER2)-negative breast cancer is the most prevalent subtype of breast cancer, which accounts for approximately 70% of all breast cancer (*Parker et al., 2009*). Neoadjuvant chemotherapy (NCT) is recommended as the first choice for patients with HR-positive, HER2-negative breast cancer, who are candidates for preoperative therapy, according to National Comprehensive Cancer Network (NCCN) guidelines (*Gradishar et al., 2022*). Neoadjuvant endocrine therapy (NET) is an acceptable, less toxic alternative to NCT. However, less than 10% of patients with HR-positive, HER2-negative breast cancer can achieve pathological complete response (pCR), and only 60% of patients respond to NCT or NET (*Sella et al., 2021*). Hence, more promising alternatives are urgently needed.

Cyclin-dependent kinase 4/6 (CDK4/6) inhibitors are breakthroughs in the treatment of HR-positive, HER2-negative breast cancer and have a remarkable efficacy in the advanced setting when combined with endocrine therapy (ET) (*Finn et al., 2015*; *Goetz et al., 2017*; *Hortobagyi et al., 2016*). Presently, abemaciclib has been approved for high-risk, early HR-positive breast cancer plus ET as adjuvant treatment, and its application is being explored in the neoadjuvant setting (neoMONARCH and NCT04293393). However, therapies with a combination of CDK4/6 inhibitors and NET only achieve a pCR rate of less than 5% (*Hurvitz et al., 2020*; *Johnston et al., 2019*). In the PALLET study, objective response rate (ORR) was not statistically significantly different between the combinational therapy and monotherapy (54.3% vs 49.5%, p=0.20; *Johnston et al., 2019*). Likewise, in the FELINE study, there was no statistical difference in clinical, mammographic, ultrasound, or MRI response between letrozole alone and combination of letrozole with a CDK4/6 inhibitor (*Khan et al., 2020*). Thus, it is reasonable to explore a novel neoadjuvant therapy for HR-positive, HER2-negative breast cancer.

Radiotherapy is one of the most important local methods to control malignancy by breaking DNA in tumor cells (*Cox and Swanson, 2013*). Stereotactic body radiation therapy (SBRT) is a novel and relatively safe radiotherapy technique and delivers high doses of radiation in a small number of fractions to the tumor lesion while minimizing radiation exposure to the surrounding tissues (*Chmura et al., 2021*). Currently, neoadjuvant radiotherapy can be applicable for inoperable breast cancer to allow an effective and sometimes more conservative surgery (*Cardoso et al., 2018*). Preclinical studies have indicated that CDK4/6 inhibitors can sensitize breast cancer cells to radiotherapy, possibly by inhibition of DNA damage repair response, enhancement of apoptosis and blockage of cell cycle progression, and inhibition of intra-tumor cellular metabolism and angiogenesis (*Han et al., 2013*; *Pesch et al., 2020*; *Petroni et al., 2021*). A preclinical study has explored different schedules of combination of radiotherapy and CDK4/6 inhibitors in human and mouse models and demonstrated that radiotherapy followed by CDK4/6 inhibitors can enhance antineoplastic effects, compared with their monotherapy alone or CDK4/6 inhibitors concurrent with or followed by radiotherapy (*Petroni et al., 2021*). Moreover, the combination of CDK4/6 inhibitors and radiotherapy can lead to a satisfying overall disease control and safety based on several retrospective studies in metastatic breast

cancer (*Bosacki et al., 2021*). Nevertheless, the efficacy and safety of this regimen in neoadjuvant treatment of early or locally advanced HR-positive, HER2-negative breast cancer are unknown.

We have conducted a prospective pilot study to determine the efficacy and safety of neoadjuvant radiotherapy followed by dalpiciclib and exemestane in patients with newly diagnosed early or locally advanced HR-positive, HER2-negative breast cancer. Dalpiciclib (SHR6390) is a potent selective inhibitor of CDK4/6 and exerts synergistic antitumor effect when combined with endocrine therapy for advanced luminal breast cancer in the DAWNA-1 trials (*Xu et al., 2021*).

## Methods

### Study design

This single-arm, non-controlled prospective pilot study in women with HR-positive, HER2-negative early or locally advanced breast cancer was registered with ClinicalTrials.gov (NCT05132790).

The study was approved by the Institutional Review Board and Ethics Committee of Shengjing Hospital of China Medical University, according to the Declaration of Helsinki and Good Clinical Practice guidelines. Written informed consent was obtained from each patient.

### Participants

Eligible patients included pre- and post-menopausal women aged between 18 and 75 years, with histologically confirmed HR-positive, HER2-negative (0 or 1+by immunohistochemistry [IHC], or 2+by IHC with negative results of fluorescence in situ hybridization), early or locally advanced invasive breast cancer with the lesion of ≥2 cm by MRI. Other main inclusion criteria were Eastern Cooperative Oncology Group performance status 0–1, adequate marrow, hepatic, and renal function. The key exclusion criteria included stage IV tumors, inflammatory breast cancer, prior systemic treatment of the current cancer, other malignancies, cardiac disease or history of cardiac dysfunction, pregnancy, lactation, and refusal to use contraception.

### Procedures

SBRT was performed every other day in the breast target lesion with a total radiation dose of 24 Gy separated by three fractions. Within 2–4 weeks after the completion of radiotherapy, individual patients received six cycles of a combination of oral 150 mg Dalpiciclib daily on days 1–21 of each 4-week cycle and 25 mg exemestane with a total duration of 24 weeks. Patients without menopause were also injected subcutaneously with 3.6 mg goserelin every 4 weeks. All patients received study treatment until completion of all prescribed protocol therapy, disease progression, unacceptable toxicity, or withdrawal of consent. If disease progression occurred, the patient was either to proceed to surgery or receive alternative neoadjuvant therapy. Surgery was performed 3–5 weeks after the last dose of dalpiciclib to allow adverse event (AE) recovery. Recommended surgery and adjuvant therapy followed per local guidelines or institutional standards. The necessity of additional chemotherapy for patients with pCR was judged by investigators. A 5-year follow-up is conducted every 3 months during the first 2 years and every 6 months for the subsequent 3 years. Additionally, safety data are collected within 90 days after surgery for subjects who discontinue study treatment.

### Outcomes

The primary endpoint was the rate of residual cancer burden (RCB) 0-I. Secondary endpoints included ORR (defined as the proportion of patients with complete or partial response according to the Response Evaluation Criteria In Solid Tumors [RECIST] version 1.1 by MRI), pCR rate in the breast and axillary lymph nodes (tpCR; ypT0/is ypN0) and in the breast (bpCR; ypT0/is ypNx), breast-conservation rate, preoperative endocrine prognostic index (PEPI) score for breast cancer-specific survival (BCSS; PEPI 0), and safety, according to the Common Terminology Criteria for Adverse Events (CTCAE) version 5.0.

### Patient-Reported Outcome (PRO) assessments

In the study, the patient-reported outcome (PRO) assessments were measured using the European Organization for Research and Treatment of Cancer (EORTC) Quality of Life Questionnaire Core-30 (QLQ-C30; version 3; *Cull, 1998*) at baseline, at the end of cycles 1–6, and before surgery. The PRO

data were scored, according to the EORTC scoring manual. A raw score was calculated as the average of items contributing to a scale and standardized to a range of 0–100 points. On global health status and functioning scales, a higher score indicated better status. Higher scores on symptom scales indicated worse symptoms.

### Tumor-infiltrating lymphocytes (TILs)

Tumor samples were collected when performing needle biopsy and surgery. The specimens were fixed in 10% formalin and paraffin-embedded. The tissue sections (3 μm) were stained with hematoxylin and eosin (H&E), and the frequency of TILs was quantified manually by a pathologist in a blinded manner. The TILs scores were determined as the percentage of TILs in tumor area (tumor cells and stroma). The percentage of stromal TILs (sTILs) was calculated in accordance with the recommendations by *Salgado et al., 2015* (*Salgado et al., 2015*).

### Programmed cell death-ligand 1 (PD-L1) expression

The expression of programmed cell death-ligand 1 (PD-L1) was characterized by immunohistochemistry (IHC) using anti-PD-L1 (clone SP142, Spring Bioscience, USA) and the percentages of PD-L1-positive tumor cells (TC) or infiltrating inflammatory cells (IC) were quantified.

### Statistical analysis

This exploratory study involves 12 patients, with the sample size determined based on clinical considerations, not statistical factors.

All statistical analyses were conducted using SAS 9.4 (North Carolina, USA). Continuous data are presented as mean and standard deviation (SD) or mean and 95% confidence interval (CI). Categorical data are expressed as frequency and percentage. The 95% CIs of pathological complete response rate, proportion of patients with RCB-0 or RCB-I, and ORR were estimated using the Clopper-Pearson method.

## Results

### Patient characteristics

Between November 2021 and January 2022, 12 HR-positive, HER2-negative patients were screened and recruited in the trial (*Appendix 1—figure 1*). Their baseline demographic and clinical characteristics are shown in *Table 1*. They had a median age of 48.5 years (range, 36–56) and the median tumor size of 33.5 mm (range, 23–90) assessed by breast MRI.

### Patient outcomes

After the completion of study treatment, two (16.7%, 95% CI 3.5%–46.0%) out of 12 patients achieved RCB 0–I tumor response, tpCR, and PEPI 0 (*Table 2*). There were three (25.0%) patients who reached bpCR. Overall, this study led to an ORR (as assessed by breast MRI) of 91.7% (95% CI 62.5–100%, *Figure 1*). The rate of conversion from mastectomy at baseline to breast conservation at surgery was 25.0% (95% CI 8.3%–53.9%). The median Ki67 of surgical specimens for patients without pCR was 7% (range, 5–70).

### Toxicity

All patients completed the study treatment and AEs are summarized in *Table 3*. The most frequent grade 1–2 AEs were leukopenia (58.3%), hot flushes (50.0%), and lymphopenia (50.0%). There were some patients with Grade 3 AEs, including neutropenia (66.7%) and leukopenia (25.0%), but not grade 4 AEs. There were no serious AEs, no therapy-related death, and no patients with dose interruption and reduction or treatment discontinuation in this population. Furthermore, there was no patient with radiation-related dermatitis and skin hyperpigmentation in this population.

### Patient-reported outcome

All patients completed the EORTC QLQ-C30 and had available results of PRO assessment (*Figure 2*). There was no clinically significant deterioration in global health status, physical, role, emotional, cognitive, and social functioning in this population.

**Table 1.** Baseline characteristics of the patients.

| | Patients (n=12) |
|---|---|
| Median age (range), years | 48.5 (36-56) |
| ≤50 | 6 (50.0%) |
| >50 | 6 (50.0%) |
| Menopausal status | |
| Premenopausal | 6 (50.0%) |
| Postmenopausal | 6 (50.0%) |
| Tumor size | |
| T2 | 11 (91.7%) |
| T3 | 1 (8.3%) |
| Lymph node status | |
| N0 | 5 (41.7%) |
| N1 | 2 (16.7%) |
| N2 | 5 (41.7%) |
| Clinical stage | |
| IIA | 4 (33.3%) |
| IIB | 3 (25.0%) |
| IIIA | 5 (41.7%) |
| Tumor grade | |
| II | 11 (91.7%) |
| III | 1 (8.3%) |
| ER expression | |
| >10% | 12 (100.0%) |
| PgR expression | |
| <20% | 3 (25.0%) |
| ≥20% | 9 (75.0%) |
| HER2 expression | |
| 0 | 7 (58.3%) |
| 1+ | 1 (8.3%) |
| 2+, FISH | 4 (33.3%) |
| Median baseline Ki67 (range), % | 35 (15–70) |
| ≤14% | 3 (25.0%) |
| >14% | 9 (75.0%) |

Data are n (%) or median (range).

ER: estrogen receptor; PgR: progesterone receptor; HER2: human epidermal growth factor receptor 2; FISH: fluorescence in situ hybridization.

## Alteration in tumor immune microenvironment

To examine the impact of this new neoadjuvant therapy on tumor environment, we performed H&E staining of tissue sections in 10 evaluable patients (*Figure 3*). Compared with those in pre-treatment biopsied tissues, treatment significantly increased the percentages of tumor-infiltrating lymphocytes (TILs) in the tumor environment (22.9% vs 10.7%, p=0.043). IHC staining of PD-L1 revealed that there

**Table 2.** Pathological and clinical response (n=12).

|  | n (%) |
| --- | --- |
| Total pathological complete response | 2 (16.7%) |
| Breast pathological complete response | 3 (25.0%) |
| Residual cancer burden score |  |
| 0 | 2 (16.7%) |
| I | 0 |
| II | 6 (50.0%) |
| III | 4 (33.3%) |
| Radiological response |  |
| Complete response | 2 (16.7%) |
| Partial response | 9 (75.0%) |
| Stable disease | 1 (8.3%) |
| Objective response rate | 11 (91.7%) |
| Preoperative endocrine prognostic index |  |
| 0 | 2 (16.7%) |
| 1–3 | 0 |
| ≥4 | 10 (83.3%) |

Data are presented as n (%).

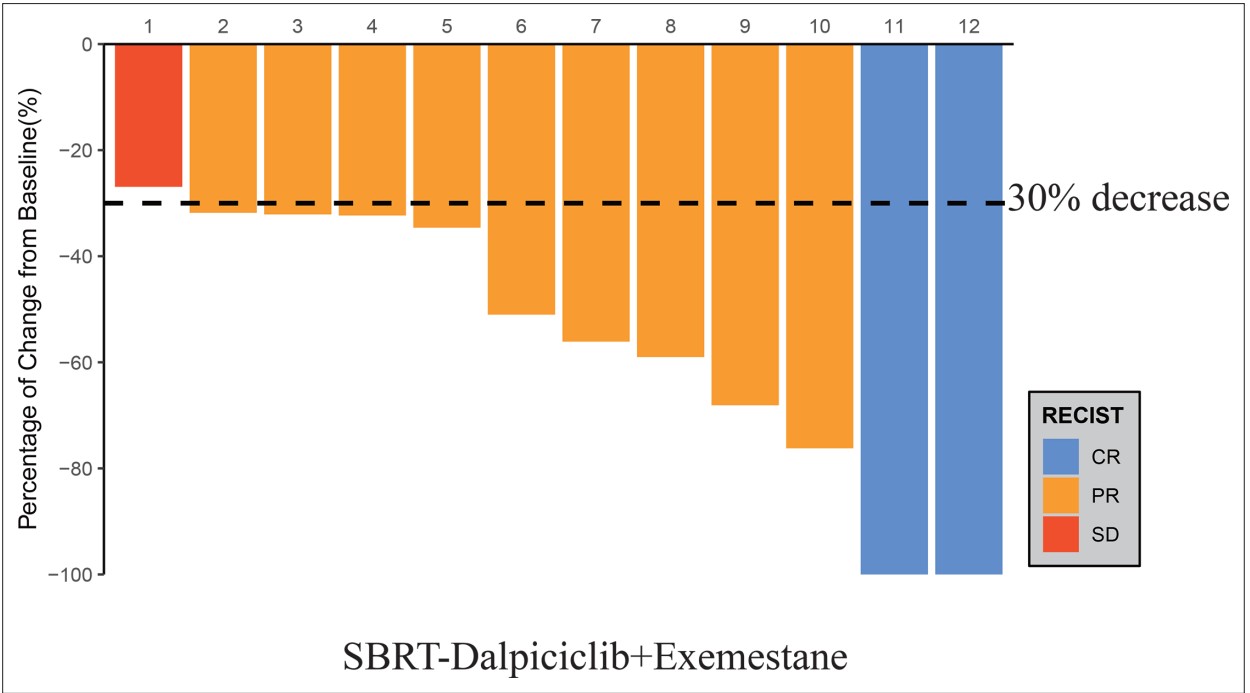

**Figure 1.** Waterfall plot of best reduction in tumor measurement from baseline.

The online version of this article includes the following source data for figure 1:

**Source data 1.** Source data for waterfall plot of best reduction in tumor measurement from baseline.

**Table 3.** Treatment-emergent adverse events (n=12).

| | n (%) | | |
| | Grade 1 or 2 | Grade 3 | Grade 4 |
|---|---|---|---|
| Neutropenia | 3 (25.0%) | 8 (66.7%) | 0 |
| Leukopenia | 7 (58.3%) | 3 (25.0%) | 0 |
| Hot flushes | 6 (50.0%) | 0 | 0 |
| Lymphopenia | 6 (50.0%) | 0 | 0 |
| Insomnia | 4 (33.3%) | 0 | 0 |
| Alopecia | 4 (33.3%) | 0 | 0 |
| Anemia | 4 (33.3%) | 0 | 0 |
| Hyperuricemia | 4 (33.3%) | 0 | 0 |
| Blood urea increased | 4 (33.3%) | 0 | 0 |
| Rash | 3 (25.0%) | 0 | 0 |
| Thrombocytopenia | 3 (25.0%) | 0 | 0 |
| Hyperglycemia | 3 (25.0%) | 0 | 0 |
| Creatinine increased | 3 (25.0%) | 0 | 0 |
| Fatigue | 3 (25.0%) | 0 | 0 |
| Arthralgia | 2 (16.7%) | 0 | 0 |
| γ-glutamyl transferase increased | 2 (16.7%) | 0 | 0 |
| Hypocalcemia | 2 (16.7%) | 0 | 0 |
| Abnormal T wave of electrocardiogram | 2 (16.7%) | 0 | 0 |
| Hyponatremia | 2 (16.7%) | 0 | 0 |
| Lactate dehydrogenase increased | 2 (16.7%) | 0 | 0 |
| Diarrhea | 1 (8.3%) | 0 | 0 |
| Headache | 1 (8.3%) | 0 | 0 |
| Mucosal inflammation Constipation | 1 (8.3%) | 0 | 0 |
| Proteinuria | 1 (8.3%) | 0 | 0 |
| Hypertriglyceridemia | 1 (8.3%) | 0 | 0 |
| Alanine aminotransferase increased | 1 (8.3%) | 0 | 0 |

Data are presented as n (%).

appeared to be no significant difference in the percent of PD-L1-positive immune cells and tumor cells between the pre-treatment and post-treatment specimens.

## Discussion

This pioneering study reported the efficacy and safety of sequential neoadjuvant radiotherapy and combination of dalpiciclib and exemestane in patients with HR-positive, HER2-negative breast cancer. This new therapeutic regimen achieved considerable anti-tumor activity with the RCB 0-I rate of 16.7% and ORR rate of 91.7%, respectively. More importantly, this combinatory therapeutic strategy was well tolerated in this population.

CDK4/6 inhibitors have recently been preclinically identified as radiosensitizers, acting mainly through hindering DNA damage repair response after radiation exposure, and enhancing cell apoptosis and blocking cell cycle, which provide a conceptual basis of combining radiotherapy with a CDK4/6 inhibitor (*Whittaker et al., 2017*; *Xie et al., 2019*). Moreover, both radiotherapy and CDK4/6

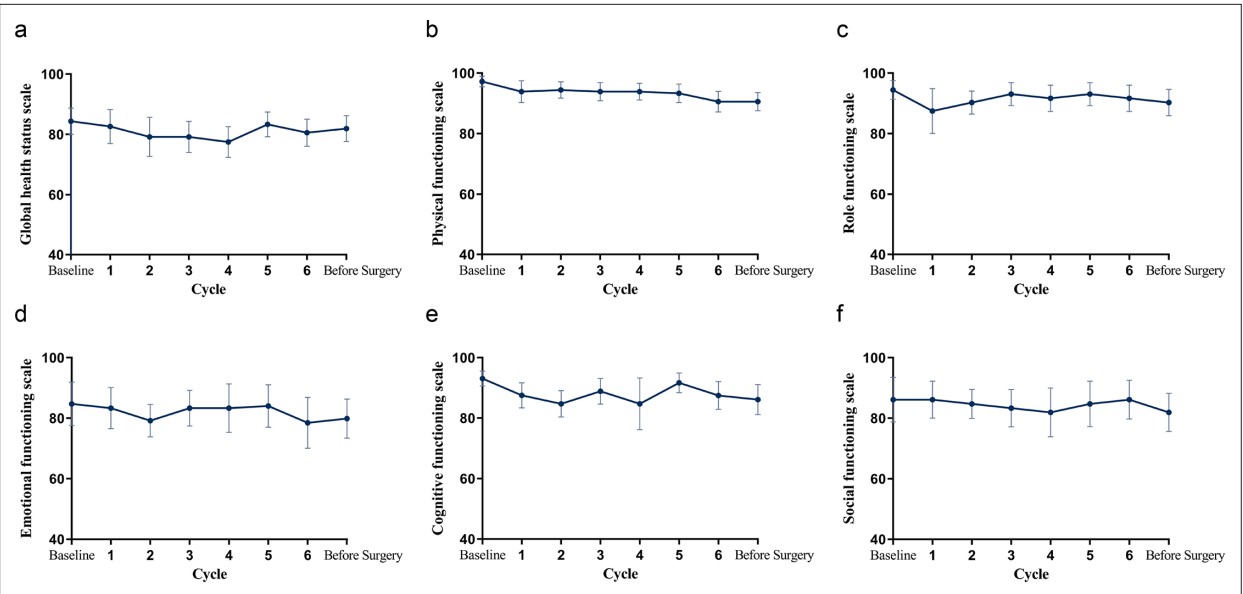

**Figure 2.** Mean change from baseline in QLQ-C30 over time. (**a**) Global health status. (**b–f**) Physical, role, emotional, cognitive, and social functioning.

The online version of this article includes the following source data for figure 2:

**Source data 1.** Source data for mean change from baseline in QLQ-C30 over time.

inhibitors exert immunomodulatory effects and may synergize anticancer immune responses (*Pesch et al., 2020*). SBRT with hypofractionation and higher radiation doses is more effective than conventional fractionation in triggering immune responses (*Gandhi et al., 2015*). Consistent with the basic research (*Petroni et al., 2021*), we used the sequential treatment with SBRT and combination of dalpiciclib and exemestane.

Unlike HER2-positive and triple-negative breast cancer, NCT or NET for HR-positive, HER2-negative breast cancer only achieves a pCR rate of ≤10% (*Cortazar et al., 2014*). Our study indicated that this novel therapeutic regimen achieved a pCR and RCB 0-I rate of 16.7%, while neoadjuvant therapies with combination of a CDK4/6 inhibitor and letrozole led to a pCR rate of 0–3.8% and RCB 0-I rate of 6.1–7.7% and treatment with NCT resulted in a pCR rate of 5.8–5.9% and RCB 0-I rate of 11.8–15.7% in HR-positive, HER2-negative breast cancer patients (*Ma et al., 2017*; *Prat et al., 2020*). It is notable that pCR rate (commonly defined as ypT0/isN0) is not significantly associated with the prognosis of HR-positive, HER2-negative breast cancer (*Ma et al., 2017*). In addition, clinical trials to test different therapies for patients with HR-positive, HER2-negative breast cancer have focused on clinical response rates as important clinical endpoints (*Hurvitz et al., 2020*; *Johnston et al., 2019*; *Khan et al., 2020*; *Whittaker et al., 2017*; *Ma et al., 2017*). In our study, treatment with the new therapeutic regimen led to 11 (91.7%) patients achieving objective responses by MRI, while previous studies have revealed that NET achieves an objective response rate of approximately 60% in HR-positive, HER2-negative breast cancer patients (*Sella et al., 2021*). One of the principal objectives of neoadjuvant therapy is to facilitate downstaging and enhance the rate of breast conservation. With the addition of SBRT, a slightly higher rate of patients changed from mastectomy to breast conservation compared with the PALLET study (25.0% vs 14.1%; *Johnston et al., 2019*). Consequently, our new neoadjuvant therapeutic regimen appeared to improve the likelihood of pathological downstaging and achieve a margin-free resection, particularly for those with locally advanced and high-risk breast cancer.

Considering that treatment with radiotherapy or a CDK4/6 inhibitor can enhance anti-tumor immunity in vitro, we analyzed the impact of new neoadjuvant therapeutic regimen on alternation in biomarker expression in the numbers of TILs before and after the treatment. Our study showed a modest but statistically significant increase in the tumor tissues with the addition of SBRT. Apparently, sequential neoadjuvant therapies with radiotherapy, a CDK 4/6 inhibitor, and exemestane turned non-responsive 'cold' tumors into responsive 'hot' ones. Therefore, our findings may provide significant

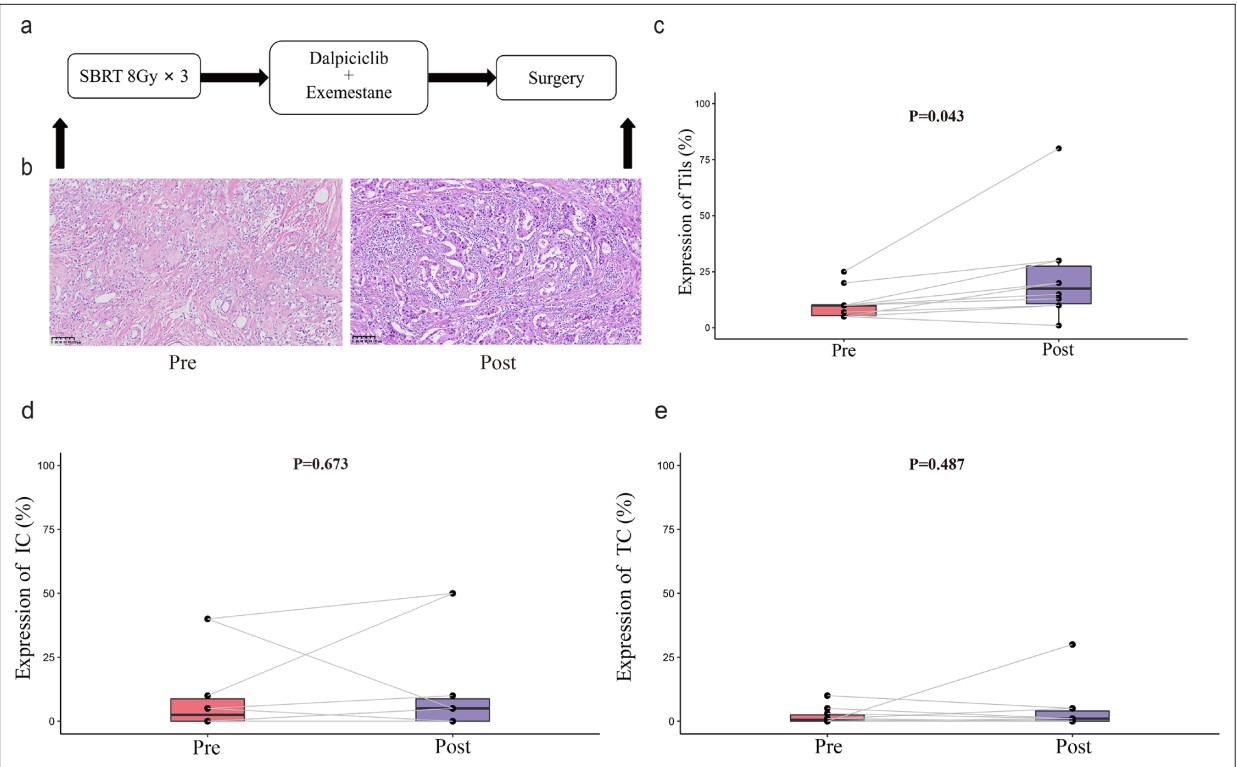

**Figure 3.** TILs and PD-L1 changes before and after treatment. (**a**) Study schema. (**b**) Representative TILs. Scale bar, 100 μM. (**c–e**) Percentage of TILs, PD-L1 expression in immune cell and tumor cell in paired specimens (n=10, pre- and post-treatment). Bars, boxes, and whiskers represent median, interquartile range, and range, respectively. Statistical analysis performed using paired t-test, statistical significance was defined as p<0.05.

The online version of this article includes the following source data for figure 3:

**Source data 1.** Source data for percentage of TILs in paired specimens.

**Source data 2.** Source data for PD-L1 expression in immune cell in paired specimens.

**Source data 3.** Source data for PD-L1 expression in tumor cell in paired specimens.

insights into designing effective therapeutic strategies against HR-positive, HER2-negative breast cancer.

Patients with HR-positive, HER2-negative breast cancer were well-tolerated to the combinational regimen. During the observation period, those patients only developed grade 1–3 AEs, including grade 3 neutropenia (66.7%), similar to that of the combination of dalpiciclib and fulvestrant in the DAWNA-1 trial (84.2%) (*Xu et al., 2021*). There were no new AEs following therapies with radiotherapy and dalpiciclib in this population.

We recognized that our study had limitations, including a one-arm preliminary exploratory trial with small sample size. The small sample size limited the comparison of our data with historical data in the literature due to the potential bias. Consequently, cross-trial comparisons like this hold limited significance. Thus, further prospective randomized clinical trials with a larger population for longer treatment duration are warranted to validate the findings.

In conclusion, our study provides novel insights into the neoadjuvant treatment of HR+/HER2- breast cancer. The sequential radiotherapy and a CDK4/6 inhibitor plus ET were effective and well-tolerated, which could serve as a feasible option for neoadjuvant therapy for HR-positive, HER2-negative breast cancers. Further validation of these findings is warranted in a large-scale study.

## Acknowledgements

Jiangsu Hengrui Pharmaceuticals provided the study drug dalpiciclib free of charge for patients enrolled in the study. We thank the patients and their families involved in this study. This research

did not receive any specific grant from funding agencies in the public, commercial, or not-for-profit sectors.

## Additional information

### Competing interests
Jianfei Wang: is an employee of Jiangsu Hengrui Pharmaceuticals Co., Ltd. Yongliang Yang: Reviewing editor, eLife. Caigang Liu: Senior editor, eLife. The other authors declare that no competing interests exist.

### Funding
No external funding was received for this work.

### Author contributions
Yu Zhang, Shuo Cao, Data curation, Formal analysis, Methodology, Writing – original draft; Nan Niu, Resources, Data curation, Formal analysis, Validation, Methodology, Writing – original draft, Writing – review and editing; Huilian Shan, Guanglei Chen, Data curation, Methodology; Jinqi Xue, Resources, Data curation, Methodology; Yongqing Xu, Jianqiao Yin, Resources, Methodology; Chao Liu, Qianshi Xu, Qingfu Zhang, Formal analysis, Methodology; Lisha Sun, Mingxuan Jia, Xu Zhang, Zhenyong Zhang, Jianfei Wang, Methodology; Xiaofan Jiang, Writing – original draft; Meiyue Tang, Resources, Data curation; Ailin Li, Conceptualization, Methodology; Yongliang Yang, Conceptualization, Methodology, Writing – review and editing; Caigang Liu, Conceptualization, Supervision, Funding acquisition, Validation, Investigation, Project administration, Writing – review and editing

### Author ORCIDs
Nan Niu ⬚ https://orcid.org/0000-0002-8206-941X
Yongliang Yang ⬚ https://orcid.org/0000-0003-0449-0599
Caigang Liu ⬚ https://orcid.org/0000-0003-2083-235X

### Ethics
Clinical trial registration NCT05132790.
Human subjects: The study was approved by the Institutional Review Board and Ethics Committee of Shengjing Hospital of China Medical University (approval identifier: 2021PS021T), according to the Declaration of Helsinki and Good Clinical Practice guidelines. Written informed consent was obtained from each patient.

Reviewer #1 (Public review): https://doi.org/10.7554/eLife.101583.3.sa1
Author response https://doi.org/10.7554/eLife.101583.3.sa2

## Additional files

### Supplementary files
MDAR checklist

Reporting standard 1. STROBE checklist.

### Data availability
The raw clinical and imaging data are protected under patient privacy laws. The datasets generated and/or analyzed during the current study are available from the corresponding author, Caigang Liu (liucg@sj-hospital.org), upon request for up to 10 years. De-identified clinical and experimental data may be shared upon request, subject to approval by the Institutional Ethical Committee(s). De-identified data will be transferred to the requesting investigator via secure file transfer. These data are permitted for academic use only and must comply with the regulations of the Human Genetics Resources Administration of China and other applicable jurisdictions. A signed data access agreement

with the principal investigator is required prior to conducting academic research. Source data files for Figures 1, 2, 3c, 3d, and 3e have been provided.

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

## Appendix 1

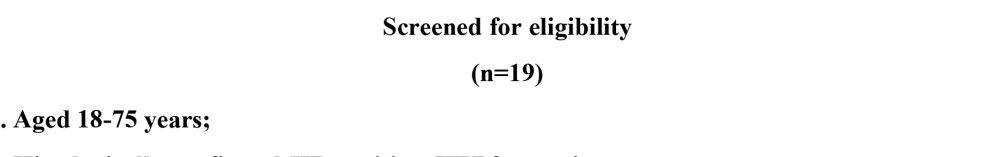

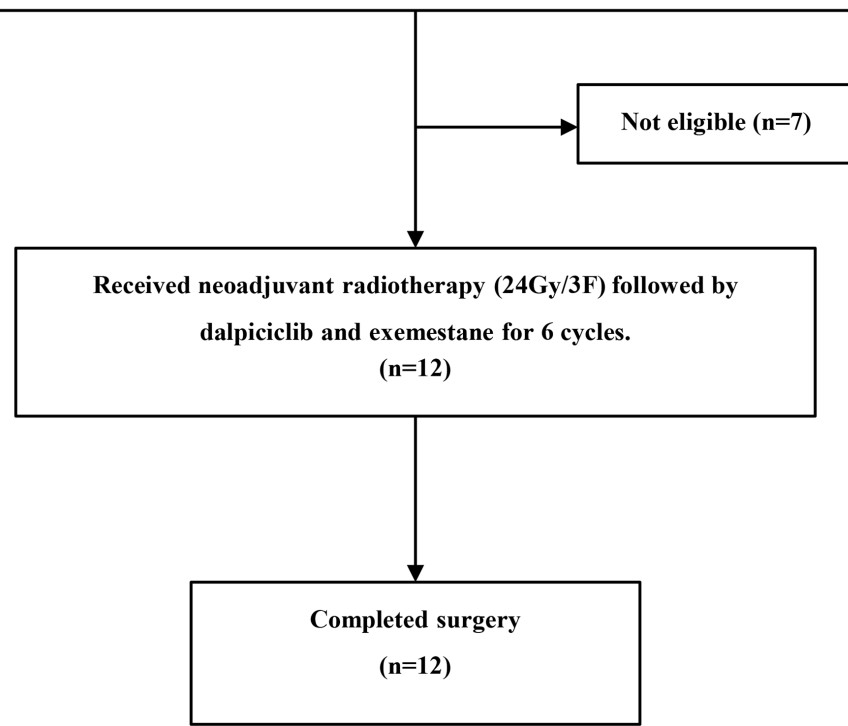

**Appendix 1—figure 1.** Flowchart of the trial.

