## [Editor Report · eLife Assessment]

The study evaluates the feasibility, safety, and tolerability of neoadjuvant radiotherapy followed by a CDK4/6 inhibitor (dalpiciclib) and hormonal therapy in treatment-naive patients with unilateral early-stage HR+/HER2- breast cancer. The findings are **convincing**, with a strong scientific rationale supported by integrated correlative studies. The trial is considered to be **important** as the outcomes could inform the design of larger, future studies. The limitations of the study have been acknowledged and outlined in this manuscript, which include only a small cohort of patients (n=12), which was not adequately powered to definitively assess the efficacy or safety of this combinatorial treatment approach.

---

## [Referee Report · Reviewer #1 (Public review)]

Summary:

This manuscript details the results of a small pilot study of neoadjuvant radiotherapy followed by combination treatment with hormone therapy and dalpiciclib for early stage HR+/HER2-negative breast cancer.

Strengths:

The strengths of the manuscript include the scientific rationale behind the approach, and the inclusion of some simple translational studies.

Weaknesses:

The main weakness of the manuscript is that a study this small is not powered to fully characterize efficacy or safety of a treatment approach, and can, at best, can demonstrate feasibility. These data need validation in a larger cohort before they can have any implications for clinical practice, and the treatment approach outlined should not yet be considered a true alternative to standard evidence-based approaches.

I would urge the readers exercise caution when comparing results of this 12-patient pilot study to historical studies, many of which were much larger, and had different treatment protocols and baseline patient characteristics. Cross-trial comparisons like this are prone to mislead, even when comparing well powered studies. With such a small sample size, the risk of statistical error is very high, and comparisons like this have little meaning.

---

## [Author Response]

The following is the authors’ response to the original reviews

**Reviewer #1(Public review):**
Summary:This manuscript details the results of a small pilot study of neoadjuvant radiotherapy followed by combination treatment with hormone therapy and dalpiciclib for early-stage HR+/HER2-negative breast cancer.Strengths:The strengths of the manuscript include the scientific rationale behind the approach and the inclusion of some simple translational studies.Weaknesses:The main weakness of the manuscript is that overly strong conclusions are made by the authors based on a very small study of twelve patients. A study this small is not powered to fully characterize the efficacy or safety of a treatment approach, and can, at best, demonstrate feasibility. These data need validation in a larger cohort before they can have any implications for clinical practice, and the treatment approach outlined should not yet be considered a true alternative to standard evidence-based approaches.I would urge the authors and readers to exercise caution when comparing results of this 12-patient pilot study to historical studies, many of which were much larger, and had different treatment protocols and baseline patient characteristics. Cross-trial comparisons like this are prone to mislead, even when comparing well powered studies. With such a small sample size, the risk of statistical error is very high, and comparisons like this have little meaning.

We greatly appreciate your evaluation of our study and fully agree with the limitations you have pointed out. We have clearly stated the limitations of the small sample size and emphasized the need for a larger population to validate our preliminary findings in the discussion section (Lines 311-316).

We acknowledge that this small sample size is not powered to characterize this regimen as a promising alternative regimen in the treatment of patients with HR-positive, HER2-negative breast cancer. Therefore, we have revised the description of this regimen to serve as a feasible option for neoadjuvant therapy in HR-positive, HER2-negative breast cancers both in the discussion (Lines 317-320) and the abstract (Lines 71-72).

We agree with you that cross-trial comparisons should be approached with caution due to differences in study designs and patient populations. In our discussion section, we acknowledge that small sample size limited the comparison of our data with historical data in the literature due to the potential bias (Lines 312-313). We clearly state that such comparisons hold limited significance (Lines 313-314) and suggest a larger population to validate our preliminary findings.

• Why was dalpiciclib chosen, as opposed to another CDK4/6 inhibitor?

Thank you for your comments. The rationale for selecting dalpiciclib over other CDK4/6 inhibitors in our study is primarily based on the following considerations:

(1) Clinical Efficacy: In several clinical trials, including DAWNA-1 and DAWNA-2, the combination of dalpiciclib with endocrine therapies such as fulvestrant, letrozole, or anastrozole has been shown to significantly extend the progression-free survival (PFS) in patients with hormone receptor-positive, HER2-negative advanced breast cancer [1-2].

(2) Tolerability and Management of Adverse Reactions: The primary adverse reactions associated with dalpiciclib are neutropenia, leukopenia, and anemia. Despite these potential side effects, the majority of patients are able to tolerate them, and with proper monitoring and management, these reactions can be effectively mitigated [1-2].

(3) Comparable pharmacodynamic with other CDK4/6 inhibitors: The combination of CDK4/6 inhibitors, including palbociclib, ribociclib, and abemaciclib, with aromatase inhibitors has demonstrated an enhanced ability to suppress tumor proliferation and increase the rate of clinical response in neoadjuvant therapy for HR-positive, HER2-negative breast cancer [3-5]. Furthermore, preclinical studies have shown that dalpiciclib has comparable in vivo and in vitro pharmacodynamic activity to palbociclib, suggesting its potential effectiveness in similar treatment regimens [6].

(4) Accessibility and Regulatory Approval: Dalpiciclib has gained marketing approval in China on December 31, 2021, which facilitates the accessibility of this medication, making it a more convenient option when considering treatment plans.

References:

(1) Zhang P, Zhang Q, Tong Z, et al. Dalpiciclib plus letrozole or anastrozole versus placebo plus letrozole or anastrozole as first-line treatment in patients with hormone receptor-positive, HER2-negative advanced breast cancer (DAWNA-2): a multicentre, randomised, double-blind, placebo-controlled, phase 3 trial[J]. The Lancet Oncology, 2023, 24(6): 646-657.

(2) Xu B, Zhang Q, Zhang P, et al. Dalpiciclib or placebo plus fulvestrant in hormone receptor-positive and HER2-negative advanced breast cancer: a randomized, phase 3 trial[J]. Nature medicine, 2021, 27(11): 1904-1909.

(3) Hurvitz S A, Martin M, Press M F, et al. Potent cell-cycle inhibition and upregulation of immune response with abemaciclib and anastrozole in neoMONARCH, phase II neoadjuvant study in HR+/HER2− breast cancer[J]. Clinical Cancer Research, 2020, 26(3): 566-580.

(4) Prat A, Saura C, Pascual T, et al. Ribociclib plus letrozole versus chemotherapy for postmenopausal women with hormone receptor-positive, HER2-negative, luminal B breast cancer (CORALLEEN): an open-label, multicentre, randomised, phase 2 trial[J]. The lancet oncology, 2020, 21(1): 33-43.

(5) Ma C X, Gao F, Luo J, et al. NeoPalAna: neoadjuvant palbociclib, a cyclin-dependent kinase 4/6 inhibitor, and anastrozole for clinical stage 2 or 3 estrogen receptor–positive breast cancer[J]. Clinical Cancer Research, 2017, 23(15): 4055-4065.

(6) Long F, He Y, Fu H, et al. Preclinical characterization of SHR6390, a novel CDK 4/6 inhibitor, in vitro and in human tumor xenograft models[J]. Cancer science, 2019, 110(4): 1420-1430.

• The eligibility criteria are not consistent throughout the manuscript, sometimes saying early breast cancer, other times saying stage II/III by MRI criteria.

Thank you for pointing out the inconsistencies in the description of the eligibility criteria in our manuscript. We deeply apologize for any confusion caused by these inconsistencies. We have revised the term from “early-stage HR-positive, HER2-negative breast cancer” to “early or locally advanced HR-positive, HER2-negative breast cancer” (Lines 128 and 150). The term “early or locally advanced” encompasses two different stages of breast cancer, whereas “Stage II/III by MRI criteria” refers to specific stages within the TNM staging system.

• The authors should emphasize the 25% rate of conversion from mastectomy to breast conservation and also report the type and nature of axillary lymph node surgery performed. As the authors note in the discussion section, rates of pathologic complete response/RCB scores are less prognostic for hormone-receptor-positive breast cancer than other subtypes, so one of the main rationales for neoadjuvant medical therapy is for surgical downstaging. This is a clinically relevant outcome.

We appreciate your constructive comments. Based on your suggestions, we have made the following revisions and additions to the article.

The breast conservation rate serves as a secondary endpoint in our study (Line 62 and 179). We have highlighted the significant 25% conversion rate from mastectomy to breast conservation in both the results (Lines 229-230) and discussion sections (Lines 290-292).

In our study, all patients underwent lymph node surgery, including sentinel lymph node biopsy or axillary lymph node dissection. Among them, 58.3% of patients (7/12) underwent sentinel lymph node biopsies.

We agree with your point that the prognostic value of pathologic complete response/RCB score is lower for hormone receptor-positive breast cancer compared to other subtypes, we have revised the discussion section to clarify that one of the principal objectives for neoadjuvant therapy in this patient population is to facilitate downstaging and enhance the rate of breast conservation (Lines 289-290). And also emphasized that this neoadjuvant therapeutic regiment appeared to improve the likelihood of pathological downstaging and achieve a margin-free resection, particularly for those with locally advanced and high-risk breast cancer (Lines 293-295).

**Reviewer #2 (Public review):**
Firstly, as this is a single-arm preliminary study, we are curious about the order of radiotherapy and the endocrine therapy. Besides, considering the radiotherapy, we also concern about the recovery of the wound after the surgery and whether related data were collected.

Thanks for the comments. The treatment sequence in this study is to first administer radiotherapy, followed by endocrine therapy. A meta-analysis has indicated that concurrent radiotherapy with endocrine therapy does not significantly impact the incidence of radiation-induced toxicity or survival rates compared to a sequential approach [1]. In light of preclinical research suggesting enhanced therapeutic efficacy when radiotherapy is delivered prior to CDK4/6 inhibitors, we have opted to administer radiotherapy before the combination therapy of CDK4/6 inhibitors and hormone therapy [2].

In our study, we collected data on surgical wound recovery. All 12 patients had Class I incisions, which healed by primary intention. The wounds exhibited no signs of redness, swelling, exudate, or fat necrosis.

References:

(1) Li Y F, Chang L, Li W H, et al. Radiotherapy concurrent versus sequential with endocrine therapy in breast cancer: A meta-analysis[J]. The Breast, 2016, 27: 93-98.

(2) Petroni G, Buqué A, Yamazaki T, et al. Radiotherapy delivered before CDK4/6 inhibitors mediates superior therapeutic effects in ER+ breast cancer[J]. Clinical Cancer Research, 2021, 27(7): 1855-1863.

Secondly, in the methodology, please describe the sample size estimation of this study and follow up details.

Thanks for pointing out this crucial omission. Sample size estimation for this study and follow-up details have been added in the methodology section. The section on sample size estimation has been revised to state in Statistical analysis: “This exploratory study involves 12 patients, with the sample size determined based on clinical considerations, not statistical factors (Lines 210-211).” The section on follow up has been revised to state in Procedures section “A 5-year follow-up is conducted every 3 months during the first 2 years, and every 6 months for the subsequent 3 years. Additionally, safety data are collected within 90 days after surgery for subjects who discontinue study treatment (Lines 169-172).”

Thirdly, in Table 1, the item HER2 expression, it's better to categorise HER2 into 0, 1+, 2+ and FISH-.

Thank you very much for pointing out this issue. The item HER2 expression in Table 1 has been revised from “negative, 1+, 2+ and FISH-” to “0, 1+, 2+ and FISH-”.